# Precision Medicine for NSCLC in the Era of Immunotherapy: New Biomarkers to Select the Most Suitable Treatment or the Most Suitable Patient

**DOI:** 10.3390/cancers12051125

**Published:** 2020-04-30

**Authors:** Giovanni Rossi, Alessandro Russo, Marco Tagliamento, Alessandro Tuzi, Olga Nigro, Giacomo Vallome, Claudio Sini, Massimiliano Grassi, Maria Giovanna Dal Bello, Simona Coco, Luca Longo, Lodovica Zullo, Enrica Teresa Tanda, Chiara Dellepiane, Paolo Pronzato, Carlo Genova

**Affiliations:** 1Lung Cancer Unit, Medical Oncology 2, IRCCS Ospedale Policlinico San Martino, 16132 Genova, Italy; tagliamento.marco@gmail.com (M.T.); giacomo.vallome@gmail.com (G.V.); massigrassi.mg@gmail.com (M.G.); mariagiovanna.dalbello@hsanmartino.it (M.G.D.B.); simona.coco@hsanmartino.it (S.C.); luca.longo@hsanmartino.it (L.L.); lodozullo@gmail.com (L.Z.); chiara.dellepiane@hsanmartino.it (C.D.); paolo.pronzato@hsanmartino.it (P.P.); carlo.genova@hsanmartino.it (C.G.); 2Department of Medical, Surgical and Experimental Sciences, University of Sassari, 07100 Sassari, Italy; 3Medical Oncology Unit, A.O. Papardo, 98158 Messina, Italy; alessandro-russo@alice.it; 4UO Oncologia, ASST Sette Laghi, 21100 Varese, Italy; alessandro.tuzi@asst-settelaghi.it (A.T.); nigro.olga3@gmail.com (O.N.); 5Oncologia Medica e CPDO, ASSL di Olbia-ATS Sardegna, 07026 Olbia, Italy; audiosini@tiscali.it; 6Medical Oncology 2, IRCCS Ospedale Policlinico San Martino, 16132 Genova, Italy; enrica.tanda@gmail.com

**Keywords:** NSCLC, biomarker, immune checkpoint inhibitor, tumor mutational burden, PD-L1, T-Cell clonality, POLE, PTEN inactivation, STK11

## Abstract

In recent years, the evolution of treatments has made it possible to significantly improve the outcomes of patients with non-small cell lung cancer (NSCLC). In particular, while molecular targeted therapies are effective in specific patient sub-groups, immune checkpoint inhibitors (ICIs) have greatly influenced the outcomes of a large proportion of NSCLC patients. While nivolumab activity was initially assessed irrespective of predictive biomarkers, subsequent pivotal studies involving other PD-1/PD-L1 inhibitors in pre-treated advanced NSCLC (atezolizumab within the OAK study and pembrolizumab in the Keynote 010 study) reported the first correlations between clinical outcomes and PD-L1 expression. However, PD-L1 could not be sufficient on its own to select patients who may benefit from immunotherapy. Many studies have tried to discover more precise markers that are derived from tumor tissue or from peripheral blood. This review aims to analyze any characteristics of the immunogram that could be used as a predictive biomarker for response to ICIs. Furthermore, we describe the most important genetic alteration that might predict the activity of immunotherapy.

## 1. Introduction

In recent years, the outcomes of patients with Non-Small Cell Lung Cancer (NSCLC) has been significantly improved with the introduction of Immune Check-point Inhibitors (ICIs). However, the identification of patients who can receive the best benefit from ICIs is still an unmet need. In the first studies with Nivolumab, an outcome analysis based on predictive biomarkers was not performed. In 2016, the OAK study with atezolizumab and the Keynote 010 study with pembrolizumab presented the first data involving correlations between clinical outcomes (response and survival) and programmed death-ligand 1 (PD-L1) expression in tumor tissue samples [1,2,3,4].

Since then, many retrospective studies and subgroup analyses have been performed, confirming the role of PD-L1 expression as a predictive biomarker of ICI efficacy. However, evidence has shown that some patients could respond even with low or absent PD-L1 expression. Furthermore, the heterogeneity and the mutability over time of the PD-L1 expression supported the hypothesis that PD-L1 alone was not sufficient for an efficient patient selection. All these findings encouraged the research of new biomarkers. Understanding the complex interaction between the immune system and cancer molecular biology could lead to the definition of a comprehensive schema that can be useful to drive treatment in single situations. This review aims at defining the state-of-the-art of biomarker research to guide ICI therapy, deepening the current knowledge in terms of approved and experimental molecules, both tissue-derived and circulating.

## 2. Established Biomarkers

### 2.1. Tissue PD-L1

PD-1, (or CD279) is a transmembrane glycoprotein expressed in different subtypes of B and T lymphocytes, natural killer cells, monocytes, Langerhans cells and Antigen-Presenting cells (APCs) [5], that plays a central role in preventing autoimmunity. Pro-inflammatory cytokines (INF, IL4, TNF, VEGF) can induce the expression of PD-1, which in turn acts by binding programmed death-ligand 1 (PD-L1, or B7-H1 or CD274) and PD-L2 (B7-DC or CD273) that are expressed on APCs, endothelial cells, dendritic cells and macrophages [6,7]. This axis represents one of the most important mechanisms of peripheral immune tolerance and regulation of T-cell activation, and involves other members of the B7 family, such as B7-1 (CD80) and B7-2 (CD86), and their respective ligands CD28 and CTLA-4 [5,6]. PD-L1 seems to have a negative prognostic value in NSCLC, but its role is still debated [8].

Four different immunohistochemistry (IHC) antibodies have been approved for the analysis of PD-L1 in lung cancer, namely, SP124, 22C3, 28-8 and SP263; among them, the last three present the highest concordance [9,10,11].

Currently, determination of PD-L1 expression on tumor specimens is the only approved biomarker to drive ICIs therapy in clinical practice, allowing the patient selection and predicting the response to PD-1/PDL1 inhibitors [12]. However, a recently published study evaluated the role of PD-L1 as a predictive biomarker among all trials that led to Food and Drug Administration [FDA] approval of ICIs. All the positive trials between 2011 and April 2019 (45 drugs among 15 different cancer types) were considered: PD-L1 was predictive in less than the 30% and not predictive in more than 50%. Moreover, PD-L1 thresholds differed across the different trials (even those involving the same tumor type) [13]. These results put into perspective the predictive role of PD-L1 and underline its limited utility as a predictor of response to cancer immunotherapy. For example, PD-L1 expression has been proven to be associated with clinical response to second-line nivolumab (CheckMate 057 trial) for non-squamous NSCLC and with pembrolizumab (KEYNOTE-010) and atezolizumab (OAK) for squamous and non-squamous NSCLC, but was neither prognostic nor predictive of benefit from nivolumab in squamous-cell NSCLC (CheckMate 017) [1,2,3,4,10,14]. A network meta-analysis updated in 2017 investigated the predictive role of PD-L1 expression among 2015 NSCLC patients treated with anti-PD-1/PD-L1 antibodies (alone or in combination) by pooling available data derived by all the published studies [15,16]. Each of the 14 selected studies evaluated PD-L1 with a different IHC test. Overall response rate [ORR] and hazard ratio [HR] for progression-free survival (PFS) and overall survival (OS) were reported according to PD-L1 positive or negative status (as defined in every single trial). ORR was 27.6% and 12.1% among 931 PD-L1 positive and 1084 PD-L1 negative patients, respectively. The ORR was significantly higher among PD-L1 positive patients (relative ratio [RR] 2.19, 95% CI 1.63–2.94; *p* < 0.01), irrespectively of the tumor histology and the ICI line of treatment. ORR increased proportionally to PD-L1 expression. Restricting the analysis to the first-line, the ORR were 40% vs. 20%, respectively for the two groups (RR 1.96, 95% CI 0.99–3.90; *p* = 0.05). The PFS, reported in 6 studies (*n* = 897), was significantly different between PD-L1 positive and negative patients, favoring the first group (HR 0.69, 95% CI 0.57–0.85; *p* < 0.01). Eight trials (*n* = 1522) observed the OS, demonstrating longer survival for patients with PD-L1 expression on tumor cells (HR 0.77, 95% CI 0.67–0.89; *p* < 0.01) [16].

Considering the first-line setting of NSCLC, ICI-monotherapy is strictly driven by a PD-L1 tumor proportion score (TPS) ≥ 50%, since an advantage in OS, as compared to platinum-based chemotherapy, has been observed only in this situation. Indeed, the KEYNOTE-024 trial was planned to include only patients whose tumor harbored high PD-L1 expression while, by contrast, KEYNOTE-042 included patients with PD-L1 TPS ≥ 1%. Both trials compared pembrolizumab to platinum-based chemotherapy and met their end-points; however, an exploratory analysis of KEYNOTE-042 suggested that the advantage observed in the pembrolizumab arm was strongly driven by the proportion of patients with TPS > 50%, who represented a significant proportion of PD-L1 positive patients [17,18,19]. However, in those trials which involved combinations of ICIs and chemotherapy, the benefit in OS derived from the addition of anti PD-1/PD-L1 agents to standard chemotherapy regimens, when demonstrated, was observed regardless of PD-L1 level (KEYNOTE-189, KEYNOTE-407, IMPOWER 150) [19,20,21,22]. Table 1 reassumes the main characteristics of principal trials exploring the role of PD-L1 in selecting patients to ICIS treatment. 

Currently, PD-L1 is considered a necessary although flawed biomarker, due to several characteristics: 

Cancer treatments might modify the expression of PD-L1 [11]. This can happen after exposure to locoregional or systemic treatments, such as radiation therapy (RT) or chemotherapy. As an example, among 76 NSCLC patients evaluated during their antineoplastic treatment, PD-L1 was tested by IHC on tumor cells and compared before and after treatment: in 29 cases PD-L1 had a major change (i.e., turned from negative to ≥ 1% or ≥ 50%, and vice versa) [23].

PD-L1 has an intra-tumoral heterogeneity, reflecting in a risk of non-representativity of diagnostic biopsy on respect with the whole neoplasia [11,24]. Notably, a high discordance of PD-L1 expression was observed between surgical specimens of resected NSCLC and the corresponding diagnostic biopsy tissues [25]. A heterogeneity can be seen also among primitive tumor and metastatic sites, but this phenomenon seems less frequent [26]. 

It was uncertain if archival samples may be suitable to perform PD-L1 testing, since fixing and preserving methods might alter the quality of tissues. However, a prespecified exploratory analysis of the KEYNOTE-010 trial, comparing pembrolizumab vs. docetaxel in previously treated PD-L1 positive advanced NSCLC, reported outcomes based on PD-L1 expression in archival (*n* = 455) versus newly collected (*N* = 578) tumor samples. Among patients with TPS ≥ 1%, HR for OS was improved by the anti PD-1 compound irrespectively of the characteristics of the specimen used: HR were 0.74 (95% CI 0.59–0.93) and 0.59 (95% CI 0.48–0.73) for archival and newly collected samples, respectively. PFS HR were also similar across archival and newly collected samples: 0.82 (95% CI 0.66–1.02) and 0.83 (95% CI 0.68–1.02), respectively [27]. 

Cytological diagnostic samples to assess PD-L1 TPS were usually excluded in clinical trials. However, in clinical practice, it is often difficult to histological material, while there is a frequent use of transbronchial needle aspiration as a diagnostic procedure [11]. As an alternative, assessment of PD-L1 expression can be performed on cell blocks when histological samples are not available [28] In conclusion, PD-L1 seems to be a robust biomarker with several limitations. As we are going to say below, other, emergent biomarkers could optimize its performance.

### 2.2. Tissue TMB

The increased acquisition of somatic mutations during tumorigenesis is reportedly associated with the formation of neoantigens and subsequent development of immunogenicity; hence, it has been postulated that tumors with a higher number of somatic mutations might be more sensitive to immune checkpoint blockade. Tumor Mutational Burden (TMB) is defined as the total number of nonsynonymous mutations per coding area of a tumor genome and is calculated as mutations per DNA Megabase (Mb) [29,30,31].

The first clinical experiences in TMB calculation involved Whole Exome Sequencing (WES); however, due to the cost and time consumption of this method, novel targeted panels for TMB measurements have subsequently been investigated [32]. With regards to the use of tissue-based TMB in NSCLC, its negative prognostic role was observed in a population of patients who underwent surgery but did not receive chemotherapy or immunotherapy [33]. The relevance of tissue TMB as a potential predictor of benefit from immunotherapy was initially explored in cohorts of patients treated with single-agent immune checkpoint inhibitors (Table 2). The role of WES-assessed tissue TMB was investigated in two separate cohorts of NSCLC patients receiving pembrolizumab (a discovery cohort, *n* = 16, and a validation cohort, *n* = 18). In this study, durable clinical benefit (partial response or stable disease for more than 6 months) and PFS were significantly improved for patients with a TMB > 209 mutations/sample (the median value in the discovery cohort) in both cohorts; moreover, 73% of patients with high TMB demonstrated durable clinical benefit, compared to 13% of patients with low TMB (*p* = 0.04) [34]. In another publication, Rizvi et al. showed that high TMB assessed through targeted NGS (MSK-IMPACT) was associated with durable clinical benefit in a population of 240 NSCLC patients treated with anti-PD-1 or anti-PD-L1 agents and was independent from PD-L1 expression [35]. The CheckMate 026 trial, designed to compare single-agent nivolumab and platinum-based chemotherapy as first line for advanced NSCLC, included WES-based tissue TMB evaluation defined as low (0–100 mutations/sample), medium (100–242 mutations/sample) and high (243 mutations/sample), as an exploratory analysis,. While the study did not meet its primary endpoint (improved PFS with nivolumab in patients with PD-L1 ≥ 5%), PFS and ORR were significantly higher with nivolumab in the subgroup of patients with high TMB. No difference in terms of OS was observed, although it should be taken into account that patients with high TMB treated in the chemotherapy arm had the possibility to cross-over to nivolumab at progression [36]. Similarly, exploratory analyses of pooled data from trials involving atezolizumab in solid tumors including NSCLC suggested a potential positive predictive role of high tissue TMB, this time defined by the 75th (high) and 50th (median) percentile of each study-specific TMB [37]. More recently, in a pooled analysis of data from KEYNOTE-010 and KEYNOTE-042, which compared pembrolizumab to, respectively, docetaxel (second-line) and platinum-based chemotherapy (first-line) in PD-L1-positive NSCLC patients, Herbst et al. identified tissue TMB defined with a pre-specified exploratory cut-point of 175 mutations/exome and measured by WES as a potential predictor of outcomes for patients receiving single-agent pembrolizumab. Indeed, high TMB was associated with improved response and survival with immunotherapy, while, by contrast, it was not associated with response to chemotherapy [38].

With regards to combination regimens, the role of tissue TMB was explored in CheckMate 227; more specifically, the study was amended to include, as coprimary endpoint, the PFS comparison between ipilimumab-nivolumab and first-line chemotherapy in patients with high tissue TMB (10 mutations per Mb, as identified through previous studies [31]); notably, PFS was significantly longer with ipilimumab-nivolumab than with chemotherapy. However, when data for the other coprimary endpoint (OS in patients with PD-L1 > 1%) were made available, ipilimumab-nivolumab achieved improved outcomes in terms of OS irrespective of TMB, thus questioning the actual role of this biomarker [39,40]. Finally, tissue TMB was explored in a pooled analysis from KEYNOTE-021, KEYNOTE-189, and KEYNOTE-407, all of which were designed to explore the benefit of adding pembrolizumab to platinum-based chemotherapy (albeit the explored different histologies); similarly to the analysis performed by Herbst et al. [38], a pre-specified cut-point of 175 mutations/exome was employed. In this analysis, no significant association between tissue TMB assessed by WES and benefit from adding pembrolizumab to platinum-based chemotherapy was observed, in terms of PFS, OS, or ORR [41] The introduction of this test in routine clinical practice is challenging, as it is essential to harmonize TMB approaches to ensure comparable results between different studies. Recently, Anagnostou et al. described a TMB corrective factor which was determined on a large cohort of tumor samples from the genome cancer atlas and then validated on a second cohort of patients treated with ICIs. It was based on different levels of tumor purity. This corrective factor seems to greatly increase the predictive value of TMB in prognostication the outcome in patients treated with ICIs, suggesting that the TMB could be largely underestimated in low tumor purity samples resulting in misclassification of patients with these tumors.

In conclusion, despite an encouraging premise and promising data from single-agent PD-1 inhibitors, TMB failed to result as an effective predictor of benefit from immunotherapy, with specific reference to OS and combination regimens. To date, tissue TMB is not ready for employment in clinical practice and its use should not be encouraged out of clinical studies aiming at optimizing the use of this biomarker.

### 2.3. Blood Sample TMB and PD-L1

Liquid Biopsy (LB) is a new powerful tool that has already entered clinical practice in advanced NSCLC through cell-free DNA (cfDNA) analysis for tumor genotyping and evaluation of the mechanisms of acquired resistance to tyrosine kinase inhibitors in molecularly selected patients [42]. Compared to conventional tissue testing, LB presents several potential advantages, since it might allow a minimally invasive monitoring of drug response, a better definition of ambiguous clinical/radiological scenarios and finally, a more accurate representation of tumor heterogeneity and the dynamic changes of tumor biology under selective pressure of anticancer treatments [43,44].

In order to overcome some of the limits of PD-L1 IHC testing, such as tissue unavailability for molecular testing in up to 30% of NSCLC patients [45] and spatial/temporal heterogeneity of IHC expression [46], some studies evaluated soluble PD-L1 (sPD-L1) or PD-L1 expression in Circulating Tumor Cells (CTCs) as an alternative source for PD-L1 evaluation as well as a dynamic biomarker in patients treated with conventional treatments [47,48,49] and/or ICIs [50,51,52,53,54]. High baseline levels of sPD-L1, assessed by an Enzyme Linked Immunosorbent Assay (ELISA), is a poor prognostic factor in lung cancer and seems associated with lower response to PD-1 blockage [55]. However, changes of sPD-L1 during ICIs have been associated with conflicting results [54]. Interestingly, some data suggest that first line chemotherapy (e.g., platinum and pemetrexed) could significantly increase sPD-L1 median level from baseline, while targeted therapies do not appear to modify sPD-L1 level [47].

Notably, the low number of patients included in these studies and the variability of assays used cannot allow to draw definitive conclusions. 

Other studies have focused on the evaluation of PD-L1 expression on CTCs by immunofluorescence, reporting a variable grade of concordance with tissue PD-L1 IHC expression [49,52,53,56]. Interestingly, increase in PD-L1+ CTCs during treatment with ICIs seems associated with disease progression [52,53] and might represent a potential minimally invasive biomarker of resistance that can assist conventional radiographic assessment in those cases of uncertain progression (pseudo-progression or mixed response). Furthermore, CTCs count seems to predict the outcome of NSCLC patients treated with ICIs, since a higher baseline CTC number has been associated with shorter PFS [53,57] and OS [53,57,58]. 

In addition to CTCs, several recent studies have investigated the potential role of cfDNA analysis as a predictive biomarker in NSCLC patients undergoing Immune Checkpoint Blockage (ICB). These studies have focused either on circulating tumor DNA (ctDNA) dynamics or on plasma identification of emerging biomarkers previously assessed in tissue, as TMB. A significant decrease of circulating tumor DNA (ctDNA) during treatment with ICIs has been associated with radiographic response and improved outcomes in advanced NSCLC in multiple studies, using different methodologies and definitions of molecular response (Table 3). In addition, ctDNA dynamics have been reported to predict tumor response in advance compared with conventional radiographic methods, potentially allowing a more accurate discrimination of complex or unclear clinical scenarios such as pseudoprogression and disease stabilization [59] and might represent a useful tool in clinical trials evaluating elective discontinuation treatment in non-progressing patients after 1–2 years of ICB. 

Additionally, ctDNA analysis might represent an alternative method for TMB assessment. The main limitation for clinical implementation of tissue TMB testing in advanced NSCLC is the unavailability of sufficient archival/fresh material in a significant proportion of patients, with only 34–59% of the samples evaluable for WES or targeted NGS in recent clinical trials [60]. Recently, several studies have evaluated the predictive role of TMB on cfDNA (also known as blood TMB, bTMB) in NSCLC patients treated with ICIs. Gandara et al. first reported on the development, testing and validation of the 394-gene FoundationMedicine (FMI) bTMB assay. The researchers retrospectively analyzed plasma samples from randomized phase II study POPLAR (training set) and phase III OAK trial (validation set), identifying a bTMB ≥ 16 mutations per megabase (Mut/Mb) as clinically meaningful and technically robust cut-point in NSCLC for predicting atezolizumab benefit. High bTMB was an independent predictive biomarker for PFS and was not associated with high PD-L1 IHC expression, as previously reported in tissue [45]. This bTMB assay was prospectively evaluated in the phase II, open-label, Blood First-Line Ready Screening Trial (B-F1RST) that evaluated the efficacy and safety of first line atezolizumab monotherapy in patients with EGFR/ALK wild type locally advanced or metastatic NSCLC, regardless of PD-L1 expression. In the biomarker evaluable population (BEP) were included patients with a baseline evaluable blood sample with a maximum somatic allele frequency (MSAF) ≥ 1% and a bTMb ≥16 which was was significantly associated with higher ORR (*p* = 0.0002) and a numerically improvement in terms of PFS (4.6 vs. 3.7 months, *p* = 0.12) and OS (not evaluable vs. 13.1 months, *p* = 0.48) [61]. The prospective validation of bTMB ≥ 16 is currently ongoing in a cohort of the phase III Blood-First Assay Screening Trial (B-FAST, NCT03178552) and the results are eagerly awaited. A second bTMB assay, using the 500-gene panel Guardant OMNI, was evaluated in an exploratory analysis of the randomized phase III MYSTIC trial. A bTMB ≥ 20 Mut/Mb was significantly associated with OS and PFS benefit with durvalumab ± tremelimumab compared with platinum-based chemotherapy, with the greatest magnitude of benefit observed for patients receiving dual immune checkpoint blockage (HR 0.49 and 0.72 for OS and HR 0.53 and 0.77 for PFS with durvalumab-tremelimumab and durvalumab alone, respectively) [62]. Once again, bTMB did not correlate with PD-L1 expression levels. The predictive role of bTMB, assessed by GuardantOMNI, to pembrolizumab alone (*n* = 31) or in combination with chemotherapy (*n* = 35) was prospectively evaluated in a small single-center study. A bTMB > 16 Mut/Mb was associated with improved outcomes after first-line pembrolizumab-based therapy and the concomitant evaluation of negative predictors in LB (i.e., mutations in STK11/KEAP1/PTEN and ERBB2) increased its predictive value for both PFS (HR 0.27 and 0.18 for bTMB alone and bTMB plus negative predictors) and OS (HR 0.47 and 0.27 for bTMB alone and bTMB plus negative predictors) [63]. These results further reinforce the potential role of plasma NGS beyond tumor genotyping of oncogene addicted NSCLCs and might provide useful information in patients candidate to ICIs, including the identification of concomitant mutations that are associated with impaired outcomes to these agents. 

The concordance between these two plasma NGS assays for bTMB estimation is unclear, albeit a recent study in metastatic castration resistant prostate cancer comparing FoundationMedicine bTMB assay, GuardantOMNI and WES on tissue, showed that the two plasma based TMB assays are highly correlated and they are also both correlated with a tissue-based TMB assay for relatively high TMB samples, but lower for low/medium TMB samples, likely due to biological differences. [64]

Besides the two commercial plasma NGS platforms, recently, Wang et al. evaluated the role of a novel 150-gene NGS panel, named NCC-GP150, that was designed and virtually validated using The Cancer Genome Atlas (TGCA) database. Blood TMB significantly correlated with tissue TMB assessed by WES in a cohort of 48 NSCLC patients with matched blood and tissue samples and was then validated in an independent NSCLC cohort of 50 patients treated with PD(L)-1 inhibitors. A bTMB ≥ 6 was associated with significantly higher ORR (39.3% vs. 9.1%, *p* = 0.02) and longer PFS (HR 0.39; *p* = 0.01) [65]. Furthermore, Georgiadis et al. employed a targeted hybrid capture NGS system (Agilent SureSelect XT in-solution hybrid capture system). The custom panel, using a validated cutoff of 5 mutations, targeting the predefined regions of interest across 58 genes reported that bTMB high, before PD-1 blockage, is predictive for PFS in solid tumor (HR 0.23, *p* = 0.003). In addition, they also reported that high MSI (≥20% of loci determined to be MSI) in plasma before treatment was associated with improved PFS (HR 0.21, *p* = 0.001), with responses associated with bTMB status, since all five MSI-H patients responding to PD-1 blockage were bTMB high, whereas six of seven progressive patients with MSI-H status were bTMB low [66]. 

Collectively, bTMB represents a promising predictive biomarker for ICIs in advanced NSCLC, albeit further prospective studies are needed before its clinical implementation. In addition to bTMB estimation, cfDNA analysis using plasma NGS panels might provide additional useful information, as the concomitant presence of somatic mutations that have been associated in tissue with primary resistance to PD(L)-1-based therapies (i.e., STK11, KEAP1) [67,68] or with improved outcomes with PD-1 plus CTLA-4 blockage (i.e., ARIAD1) [68], as well as might rescue a significant proportion of oncogene-addicted NSCLC patients with incomplete molecular profiling [69] that can derive less benefit from ICB.

## 3. Tumor Infiltrating Lymphocytes (TILs in Lung Cancer)

Assessment of the Tumor Immune Microenvironment (TME) has become an interesting biological and clinical-biomarkers as a result of many data supporting the prognostic and potentially predictive significance of Tumor-Infiltrating Lymphocytes (TILs) in many different tumor types [76]. 

Overall, the presence and the quantity of TILs has been associated with improved prognosis in many different cancers [77,78,79,80]. Indeed, the localization, density and functional orientation of TME have a principal role in directing tumor development or regression. TME consists of lymphocytes (T and B cells), responsible for adaptive immunity, and myeloid cells, that participate in both the innate and adaptive immunity. All these elements communicate with each other, with stroma and with tumor cells through tumor-derived cytokines and neoantigens [81]. 

In the TME different TILs have distinct functions and different clinical impacts. Cytotoxic CD8+ are capable of killing cancer cells directly, while CD4+ are a heterogeneous class of cytokine secreting lymphocytes (Th1, Th2, Th17, and Treg CD4+) involved in the activation and the inhibition of CD8+ (respectively, IFN-γ, IL2 secreted by Th1 and IL-4, IL-5, IL-9, IL-10, IL-13, IL-25 secreted by Th2) [78,79]. CD45RO+ T-cells compose another subclass of TILs, considered as memory T lymphocytes. Regulatory T cells (Tregs) possess an immune-inhibitory function and are able to maintain immune homeostasis; these cells are regulated in development and function by the transcription factor FOXP3. 

Immunohistochemistry is the optimal method to evaluate TIL subsets. Nevertheless, several studies have investigated total TIL levels using standard Hematoxylin and Eosin (H&E) staining and found strong prognostic and predictive impact [81]. In NSCLC, several studies evaluated TILs in H&E routine slides with various scoring models [82,83,84,85] but no consensus has been reached so far. 

The prognostic significance of TILs in primary NSCLC has been investigated in numerous studies in patient with resected NSCLC. On the other hand, few studies have been published in patients with advanced NSCLC, because routine H&E staining (rather than IHC) to assess lymphocyte infiltration is difficult due to the small biopsy available in this setting [84,85]. High TIL density was associated with increased survival pathological stage I [84] and III [85]. A large Italian-cohort study including 1290 patients found survival benefit among patients with high TILs infiltration in the SCC histology, while no survival association was observed in the whole cohort. These results were in contrast with a recent large NSCLC study published by Brambilla et al., where the survival benefit was achieved in all histology subtypes [82]. 

Until now, the most robust prognostic TIL marker in NSCLC is CD8+ [86,87]. Indeed, extensive stromal infiltration by CD8+ TILs is strongly associated with patient survival. On the contrary, no conclusive results have been achieved on the prognostic impact of CD4+ TILs in NSCLC [80,88] and, among the CD4+ subsets, Th1 lymphocytes have been associated with improved survival, while Th2 lymphocytes have been associated with tumor progression [89,90]. Overexpression of CD45RO+ among TILs has been associated with improved outcomes in various cancers as well as NSCLC [91] while high infiltration of FOXP3+ Tregs has been correlated with poor survival in NSCLC [92].

Preliminary observations of patients with recurrent cancers indicate that clinical responses to immune checkpoint blockers are associated with elevated tumor levels of immune inhibitory signals, such as PD-L1, cytotoxic T-lymphocyte-associated protein 4 (CTLA-4) and with increased numbers of TILs. Tokito et al. retrospectively analyzed predictive relevance of PD-L1 expression combined with CD8+ TIL density in 74 stage III NSCLC patients receiving concurrent chemoradiotherapy. Four groups of patients were analyzed: CD8+ high/PD-L1 neg; CD8+ low/PD-L1 pos; CD8+ high/PD-L1 pos; CD8+ low/PD-L1 neg. At a median follow up of 53 months, the best outcome in term of PFS and OS was showed by CD8+ high/PD-L1 neg group (median not reached), followed by CD8+ high/PD-L1 pos group with a PFS of 17.6 months and an OS of 35.3 months. On the other hand, the group with the worst results was CD8+ low/PD-L1 pos with a PFS of 8.6 months and an OS of 13.9 months. These results underline the negative prognostic value of PD-L1 and highlight the importance of lymphocyte-tumor interface. It is known that RT increases the expression of PD-L1 and promotes anti-tumor immunity [93], therefore, it is reasonable to think that, in the era of immunotherapy, the combination of PD-L1 and CD8 expression may have an important predictive response value [94].

All the studies with PD-1 ICIs nivolumab and pembrolizumab focused on tumor PD-L1 expression as a predictive biomarker, without considering the predictive role of TILs [1,2,3]. On the contrary, studies with the PD-L1 inhibitor atezolizumab confirmed the predictive role of PD-L1 expression on tumor cells but considered also the PD-L1 expression on TILs: this analysis validated the predictive role of TILs in this setting [4].

In an original report, Roger Sun et al. developed a computerized tomography (CT) derived radiomic signature of tumor immune infiltration (CD8+ cells) and analyzed its correlation with outcomes in patients treated with immunotherapy. First, they developed and validated, in four independent cohorts for a total of 135 patients with advanced solid tumors, a radiomic signature by combining contrast-enhanced CT images and RNA-seq genomic data derived from tumor biopsies to assess CD8+ cell tumor infiltration. The genomic data were based on the *CD8B* gene that allowed to estimate the abundance of CD8 cells in the samples. The clinical validation cohort was composed by 137 patients recruited at the Gustave Roussy Hospital and treated with anti-PD-1 and anti-PD-L1 monotherapy in the MOSAICO trial. In these patients a high baseline radiomic score was associated with a higher proportion of objective response at 3 months (*p* = 0.049) and with an improved overall survival in univariate (median overall survival 24.3 months vs. 11.5 months in the low radiomic score group, *p* = 0.0081) and multivariate analyses (*p* = 0.0022). This algorithm provides a promising way to predict the immune phenotype, could be useful in estimating CD8+ cell count and predicting clinical outcomes of patients treated with immunotherapy but require to further prospective randomized trials [95].

TILs in NSCLC confirm their prognostic role and seem to be a predictive factor. Especially, the combination of CD8+ and PD-L1 expression seems to be more robust biomarker than PD-L1 alone. However, the direct evaluation of TILs remains difficult due to the small histologic material derived from the biopsy and due to the inability of the single bioptic sample to reflect the microenvironment of each metastases. Despite this, the introduction of radiomic immune infiltration signature could be a valid, non-invasive method to evaluate TILs and to define their role as a biomarker.

## 4. Emergent Biomarkers

### 4.1. T-Cell Clonality

Molecular mechanisms underlying resistance to ICIs are imperfectly understood. Recent investigations have focused on the peripheral blood T-cell receptor (TCR) repertoire. Upon recognizing antigens, antigen-reactive T cells are activated and proliferate, a process leading to clonal expansion [96]. Tumor recognition by T cells is impaired in cancer patients [97]. Nevertheless, tumor-specific T cells occur responding to tumor antigens that include individual “neoantigens” derived from mutated proteins in cancer cells [34,98,99,100]. These tumor-specific T cells, however, may remain anergic [98]. T-cell clones can be tracked by determining TCR rearrangements composed of variable (V)-diversity (D)-joining (J) region genes, which generate the antigen-specific complementarity determining region 3 (CDR3). Analysis of T-cell clonality may therefore reveal the degree of tumor-antigen driven T-cell expansions and help to dissect mechanisms underlying T-cell tolerance to cancer antigens. The reactivity of TCRs expressed by TILs determines their capacity to interact with tumor antigens presented on antigen presenting cells (APCs). Accordingly, the TCR repertoire has been reported to be associated with response to immune checkpoint blockade and survival in cancer patients [101,102]. T-cell responses against certain tumor antigens were detected before initiation of CTLA4 blockade in patients responding well to the immunotherapy [103]. Dense CD8+ T-cell infiltration in the tumor microenvironment correlated with better prognosis under PD-1 inhibition [101]. Recently, a paper by Reuben et al. has described tissue t-cell repertoire in localized NSCLC. Their findings suggested a positive relation between T cell density and clonality. Furthermore, tumor with high PD-L1 demonstrated high T cell density and clonality; in addition, TMB was correlated to high T cell clonality. Indeed, EGFR mutant NSCLC presented a lower T cell clonality, that could be a possible explanation of the lower activity of ICIs in these patients [104]. T-cell clone analyses are recently considered as useful for early diagnoses of IrAEs (Immune-related Adverse Events). It has been shown using next generation sequencing that patients who experienced severe IrAE under CTLA4 inhibitor exhibited higher numbers of T-cell clones expanding after the treatment even before the clinical symptoms of IrAEs [105,106,107]. In patients who developed severe immune-related adverse events (IrAEs), CD4^+^ and CD8+ TCR spectratypes became more restricted during anti-CTLA4 treatment, suggesting that newly expanded oligoclonal T-cell responses may contribute to IrAEs [108]. Arakawa et al. demonstrated the presence of several T-cell clones in the blood of melanoma patients prior to immunotherapy, which may reflect the extent to which T cells are able to react against melanoma and potentially control melanoma progression. Therefore, T-cell clonality in the circulation may have predictive value for antitumor responses from checkpoint inhibition [108]. Hogan et al. measured the combinatorial diversity evenness of the TCR repertoire in pre-treatment peripheral blood mononuclear cells from melanoma patients treated with anti-CTLA4 or anti-PD1. The evaluation of basal TCR repertoire diversity in peripheral blood could help to predict responses to anti-PD1 and anti-CTLA4 therapies [109]. Moreover, Postow et al. demonstrated that baseline TCR diversity in the peripheral blood was also associated with clinical outcomes in melanoma patients [110]. Miyauchi et al. showed that the low clonal T cell expansion in NSCLC with *EGFR* mutations might be a critical factor related to the unfavorable response to ICI. Furthermore, TCR sequencing might be applicable for the treatment selection in patients with *EGFR* mutations by evaluating the proportions of TCRβ clones in the tumor. Despite these insights, it is elusive whether blood T-cell repertoires have potential prognostic values or not.

### 4.2. PTEN Inactivation

Phosphatase and tensin homolog deleted on chromosome 10 (*PTEN*) is a key tumor suppressor gene mapping at chromosome 10q23 that was originally identified in 1997 [111]. PTEN protein is quite ubiquitously expressed and it exerts its function by acting as a phosphatase. Indeed, by dephosphorylating PIP3 to PIP2, PTEN eventually leads to inhibition of the PI3K/mTOR/Akt signaling pathway, a major pathway that regulates cell growth, survival, and migration [112]. Unlike other tumor suppressor genes such as Retinoblastoma Transcriptional Corepressor 1 (RB1) [113] or Adenomatous Polyposis Coli (APC) [114], which follow the Knudson’s “two hit” hypothesis for their inactivation [115], PTEN function was demonstrated to be impaired by even a modest reduction of its expression levels, thus behaving as a cancer susceptibility gene [116]. Apart from the negative regulation of the PI3K/mTOR/Akt oncogenic pathway, PTEN has also been documented to play a role in the DNA repair process and in maintaining chromosomal integrity [117].

The involvement of *PTEN* has been widely reported in lung cancer, although genetic alterations such as mutations and deletions are not the main recurring event, ranging from 2% to 7% of the cases [118]. Conversely, the loss of PTEN protein is more frequently observed, involving about 40% of lung cancer patients and showing a correlation with smoking history, squamous histotype and shorter survival [118]. Interestingly, PTEN loss of activity has also been linked to resistance to targeted treatments and to immunotherapy, as well.

In this context, a complex interplay between the regulation of oncogenic pathways, such as that exerted by PTEN, and the response to immunotherapy has been recently depicted by Peng and colleagues in melanoma pre-clinical models [119]. Indeed, resistance to T cell-mediated immunotherapy was stated as a possible consequence of the loss of PTEN, which was reported to correlate with decreased T-cell infiltration at tumor sites and with a worse outcome in melanoma patients treated with anti PD-1. Intriguingly, the efficacy of immune checkpoint inhibitors was improved in mice administered with a selective PI3Kβ inhibitor, suggesting that PTEN loss may promote immune resistance through the up-regulation of the PI3K/mTOR/Akt pathway. Combinations of immunotherapy and inhibitors of the PI3K-Akt pathway are, therefore, worth to be further investigated [119].

In 2016, the involvement of PTEN in the response to the immune checkpoint blockade therapy has also been investigated in lung cancer [120]. Here, the authors integrated data from mutational profile and tumor infiltrating cells in 113 advanced patients treated with anti CTLA-4/PD-1. Notably, *PTEN* mutations were detected in non-responders only, supporting the potential role of this tumor suppressor gene in the immune resistance also in lung cancer.

### 4.3. POLE Mutations

The DNA polymerase epsilon catalytic subunit A (*POLE*) is a broadly expressed gene that maps at chromosome 12q24.33 and is one of four subunits that form the DNA polymerase epsilon (Polε), an enzyme complex responsible of the synthesis of the leading strand during DNA replication. POLE is the major catalytic and proofreading subunit of Polε, owning replicative capability and 3′-5′ exonuclease activity. As such, this enzyme is involved in both DNA replication and DNA repair pathways, which are crucial for the high-fidelity activity needed to prevent tumorigenesis [121]. Mutations in the proofreading domain of *POLE* have been reported as pathogenic, resulting in approximately a 100-fold increase of the mutation rate, thereby increasing the tumor mutational burden (TMB). Tumors that harbor *POLE* mutations are, therefore, generally referred to as “ultramutated” [122]. Consequently, these tumors show increased neoantigen load and tumor infiltrating lymphocytes, and they may be potentially more responsive to immunotherapy [123,124].

*POLE* mutations have been identified in 7–12% of endometrial and 1–2% of colorectal cancers as well as in other tumors [125,126], including NSCLC [127,128]. Interestingly, endometrial cancers with somatic *POLE* proofreading domain mutations have an outstanding prognosis, which probably depends on the abundance of antigenic neoepitopes, able to stimulate a strong antitumor immune response [125]. However, it has been recently reported that prediction of this excellent outcome in patients with endometrial tumors should be restricted to cases harboring hotspot *POLE* mutations, whereas mutations with unknown significance should be further investigated [129]. In 2018, *POLE* mutation frequency has been evaluated in patients with NSCLC by next-generation sequencing. As a result, 9 out of 319 (2.8%) patients showed *POLE* mutation. Intriguingly, in *POLE* mutated lung adenocarcinomas the median TMB was increased by 1.6-fold per Mb (*p* = 0.026), and both PD-L1 expression and CD8+ tumor infiltrating lymphocytes were higher in this subgroup of NSCLC patients, suggesting *POLE* mutation as a candidate biomarker to predict the response to immunotherapy [127]. In parallel, Liu et al. [128] investigated a cohort of 513 patients with lung adenocarcinoma and 497 with squamous cell carcinoma from The Cancer Genome Atlas (TCGA) to test the prognostic value of *POLE* mutations and PD-L1 expression. Of patient cohorts, 6% and 5.6% were positive for a *POLE* mutation in adenocarcinoma and squamous cell carcinoma, respectively. Results indicated *POLE* mutations as a favorable biomarker for improved OS in squamous cell carcinoma patients (*p* = 0.033) only, although *POLE* mutated adenocarcinomas with high expression of PD-L1 also exhibited improved OS (*p* = 0.024). This benefit has been linked to the activation of antitumor immune response pathways rather than to the presence of the tumor infiltrating lymphocytes [128].

In summary, lung cancer patients can also harbor *POLE* mutations, which in the future might be considered in biomarker panels aimed at predicting individuals, who would benefit from immune checkpoint inhibitor therapy.

### 4.4. Lymphocyte Ratio

Several studies hypothesized that pretreatment peripheral blood cells could be associated with outcomes in NSCLC patients treated with immunotherapy, with particular regard to lymphocytes and neutrophil-to-lymphocyte ratio (NLR).

Some retrospective series [130,131,132,133,134] suggested a relationship between high baseline NLR and worse outcomes during immunotherapy with ICIs. This finding has been confirmed by some little prospective studies [135,136]. These series showed, respectively, that elevated NLR, HLA-DR^low^ monocytes and low frequency of dendritic cells were associated with shorter OS and PFS [135], and that elevated NLR and platelet-to-lymphocyte ratio (PLR) were associated with shorter OS and PFS and with lower response rate [136]. However, it remains unclear whether NLR is actually predictive or prognostic. In fact, NLR has been included in the Gustave Roussy Immune Score, a validated prognostic score for patients in immunotherapy phase I trials [137].

Moreover, none of the studies mentioned above provided for a control arm. Russo et al. [138] conducted a study enrolling 23 patients with NSCLC treated with nivolumab and 27 NSCLC patients treated with docetaxel as controls: baseline neutrophilia, thrombocytosis and high derived-NLR were associated with no response (0% ORR) to both nivolumab and docetaxel; also elevated PLR was associated with reduced response to nivolumab and no response to docetaxel.

Another trial [139] showed a relationship between clinical efficacy of atezolizumab therapy and lymphocyte ratio modifications, but no baseline value was correlated with the outcomes.

In a cohort of 31 patients receiving nivolumab, the absolute number of baseline circulating NK cells resulted 2-fold higher in clinical benefit group (partial response, complete response or stable disease) compared to non-responders group [140]; the findings were then approached by the assessment of the Receiver Operating Characteristic (ROC) curve demonstrated a high sensitivity and specificity. As seen above, the different lymphocyte ratios have the advantage of being easily calculable, minimally invasive, but have shown a prognostic meaning rather than predictive of response to ICIs. Probably, they could be included within more complex scores of prognosis evaluation as objective data.

### 4.5. Co-Mutation of KRAS e STK11

The *RAS* family genes encode for small guanosine triphosphatase (GTPase) proteins, of which K-ras, H-ras, and N-ras are the best known. Their activation leads to transmit signals to multiple biochemical pathways such as RAF-MEK-ERK, PI3K-AKT-mTOR, and RALGDS-RA that regulate cell proliferation, differentiation, motility, and apoptosis. Somatic activating mutations of the *KRAS* gene are the most observed in NSCLC, especially in lung adenocarcinoma (about 20–30%), and less frequently in in squamous cell carcinoma (about 5%) [141]. The most frequent mutations occur in codons 12 and 13, are associated with cigarette smoking and confer an adverse prognosis compared to *KRAS* wild-type [142]. Recent studies have demonstrated, through gene expression profiling, that *KRAS* mutations are usually but not always mutually exclusive with *EGFR* mutations or *EML4-ALK* rearrangement [35,143,144] and rarely can co-exist with other genomic alterations [145]. A large analysis, testing 1343 NSCLC tumor samples with NGS, showed a significant association between *EGFR* and *PI3KCA* or *CTNNB1* mutations and between *KRAS* and *STK11* mutations [146]. In particular, *STK11* gene encodes for a serine/threonine kinase 11, also known as liver kinase B1 (LKB1), that directly phosphorylates and activates an AMP-activated protein kinase (AMPK), a metabolic checkpoint known to regulate lipid, cholesterol and glucose metabolism in specialized metabolic tissues such as liver, muscle and adipose [147]. Several works have revealed that one of the major growth regulatory pathways controlled by LKB1-AMPK is the mammalian target-of-rapamycin (mTOR) pathway that controls cell growth in all eukaryotes and is deregulated in most human cancers [148,149,150]. Somatic mutations of *STK11*/*LKB1* are found in several malignancies including NSCLC where they may be present in up to 30% of the patients, more frequently in adenocarcinoma than in squamous cell carcinoma [151,152]. Furthermore, it has been demonstrated that concomitant mutations in *KRAS* and *STK11* confer poor survival in lung adenocarcinoma patients [153] but their role in response to different kinds of treatments, including immunotherapy, need to be further elucidated.

Recently, a large subgroup analysis of *KRAS* mutant NSCLC patients treated with nivolumab within the Italian Expanded Access Program clearly showed that nivolumab did not improve ORR, neither PFS nor OS in *KRAS* mutant respect to *KRAS* wild type patients (ORR, 20% vs. 17%; median PFS, 4 vs. 3 months; median OS, 11.2 vs. 10 months), although the 3-months PFS rate was significantly longer in *KRAS* mutant than *KRAS* wild type patients (53% vs. 42%) [154]. Since the inactivation of *STK11* by mutational or non-mutational mechanisms is associated with an inert tumor immune microenvironment and lower PD-L1 expression [155,156], it has been suggested that co-occuring *KRAS* and *STK11* (*KRAS*/*STK11*) alterations were associated with poorer response to ICIs for patients with NSCLC. Skoulidis and colleagues, in a retrospective analysis including 174 patients with *KRAS* mutant lung adenocarcinoma treated with nivolumab, showed significantly lower ORR among patients with *STK11* alterations than among those with *TP53* alterations (7.4% vs. 35.7%). The authors confirmed this observation comparing the ORR between the two subgroups in the CheckMate057 randomized phase 3 clinical trial (ORR = 0%, 0/6 vs. 57.1%, 4/7) and in the GEMINI trial (ORR = 0%, 0/6 vs. 53%, 9/17) [67]. Similar results were observed in a cohort of patients treated with nivolumab plus ipilimumab in the CheckMate-012 study (ORR = 0%, 0/3 for *KRAS*/*STK11* vs. 78%, 7/9 for *KRAS*/*TP53*) [157]. Also, PFS and OS were significantly shorter in patients with *KRAS*/*STK11* co-mutant tumor than in those with *KRAS* mutant and *STK11* wild type (*KRASm*/*STK11wt*) tumor (PFS: *p* < 0.001; OS: *p* = 0.0015) [67]. Lastly, even though *KRAS*/*STK11* mutant tumor have lower PD-L1 expression when compared *KRASm*/*STK11wt*, the presence of mutations resulted independent of PD-L1 expression; furthermore, the association of *STK11* mutation with worse clinical outcome was also confirmed among PD-L1 positive subgroup of *KRAS*/*STK11* mutant NSCLC patients treated with ICIs [67]. Koyama et al. demonstrated that the co-mutation of STK11 resulted in accumulation of neutrophils with T-cell–suppressive effects, and an increase in the expression of T-cell exhaustion markers and immunosuppressor cytokines in mouse model of KRAS-driven NSCLC. Furthermore, the number of TILs was also reduced. Finally, It was described a reduced expression of PD-1 ligand PD-L1 [158].

Finally, another recent work showed that NSCLC patients with *KRAS*/*STK11* co-mutation, treated with ICIs as second- or third-line of therapy, experienced a worse PFS and OS compared with those with *STK11* mutation alone (median PFS, 2.4 months vs. 5.1 months, *p* = 0.048; median OS, 7.1 months vs. 16.1 months, *p* < 0.001) [159].

Altogether, these data suggest the *STK11*/*LKB1* mutation as a major driver of immune-escape and that co-occurring *KRAS*/*LKB1* mutation may be a potential biomarker for limited clinical benefits achieved by ICIs in advanced NSCLC. However, due to a limited amount of available data, further studies are warranted to evaluate the impact of *KRAS*/*STK11* co-mutation in response to immunotherapy.

### 4.6. IFN-Gamma and JAK/STAT Axis

Many hypotheses have been made regarding the expression of interferon gamma (IFN-g) and its prognostic and predictive value of response to immunotherapy.

In cancer, IFN-g is mainly produced by TILs in the figure of T cells and NK cells. [160] This protein can up-regulate the expression of PD-L1; indeed, in a recent study including 21 melanoma patients and 17 NSCLC patients treated with Nivolumab, it has been demonstrated that an high expression of IFN-g mRNA was associated with longer PFS and OS in both groups even though the OS difference in the NSCLC patients was not statistically significant. [161]. Other similar studies involving different solid tumors, albeit limited by small sample size, confirmed that IFN-g may be a good predictive biomarker of response to anti-CTLA4 or anti-PD-1/PD-L1 ICIs [12].

The strong rationale behind this hypothesis, besides the direct up-regulation of PD-L1/L2, is based on the IFN-g downstream signaling. IFN-g binding its receptor lead to phosphorylation of JAK 1 and JAK2 and, then, to the dimerization of STAT1-2. These dimers accumulate in the nucleus where they act as transcription factors via IRF1 that finally increase the surface expression of PD-L1 [162,163].

Several clinical trials testing the expression of IFN-g with different modalities (mRNA, gene signatures, etc.) are ongoing to validate a standard method that can be feasible in the daily clinical practice.

### 4.7. IL-6

IL-6 is a pro-inflammatory cytokine produced by T-cells and macrophages usually involved in the tumor progression and immune regulation in a pathway connected to IFN-g. In contrast with IFN-g, this cytokine down-regulates the expression of surface PD-L1 and HLA class I; this may be a mechanism of tumor escape and resistance to treatment with ICIs [162].

This effect has been proved in humans by Bjoern et al. [164], who observed that in a cohort of melanoma patient onset of high levels of IL-6 during the treatment with anti-CTLA4 therapy was associated with worse prognosis a resistance to treatment.

With regards to NSCLC, a prospective study showed that high IL-6 is a prognostic factor predicting poor OS regardless of the treatment [165].

### 4.8. B7-H4

The V-set domain containing T cell activation inhibitor 1 gene (chromosome 1p13.1-p12) encodes for the protein B7-H4, a type I transmembrane protein that belongs to the B7 immunoglobulin superfamily. B7-H4 negatively modulates the T cell function. Although its receptor is still unknown, evidence indicates that it can be induced on activated T cells. Once B7-H4 binds the T cells, it triggers a strong inhibitor effect on cell proliferation, interleukin secretion, as well as cytotoxicity activity [166] [167,168]. B7-H4 has also been reported to play a relevant role in cancer development and progression by inhibiting apoptosis and accelerating the proliferation, migration and invasion of the cells [169]. Zhen-Ye Li et al. [170], evaluated B7-H4 expression in brain metastases from NSCLC, reporting that patients whose metastases were strongly positive for B7-H4 expression had a shorter median OS compared to patients with low expression of B7-H4 (11.4 months vs. 26.2 months; *p* = 0.002) [170]. In addition, consequently to the negative modulation of T cells, B7-H4 has been suggested as a predictor of response to immune checkpoint blockade. Recently Genova et al. performed a pilot study analyzing B7-H4 protein expression on a cohort of 46 advanced NSCLC patients treated with nivolumab. Notably, patients whose tumor was B7-H4 positive (cut-off >1% of tumor cells) had more than 2-fold increased risk of disease progression and tumor-related death compared to B7-H4 negative patients [171].

### 4.9. Exosomes

Exosomes are extracellular vesicles with a size ranging from 40 to 100 nm. These small vesicles, released by an endosomal maturation, carry biomolecules such as genetic material, including DNA, mRNAs, miRNAs and other noncoding RNAs, as well as proteins [172]. Exosomes can be found in various body fluids, such as the blood, sputum, urine, or malignant effusions, and they play a pivotal role in intercellular communication [173]. Interestingly, tumor-derived exosomes bearing their molecular and genetic cargo may reflect the composition of tumor cells. A growing body of evidence indicates that tumor-derived exosomes can rescue tumor cells by evading the surveillance of immune cells, which could represent a therapeutic target [174]. Tumor-derived exosomes in fact, can suppress the function of immune cells by transferring their content, thus regulating lung cancer progression [175]. Recent evidence has demonstrated that exosomes from various tumor cells, including lung cancer cells carry immunosuppressive PD-L1 on their surface, which can be up-regulated by IFN-γ, thereby facilitating tumor growth by suppressing the function of CD8+ T cells and the immune response [176,177]. In particular, the level of circulating exosomal PD-L1 positively correlates with that of IFN-γ in patients with metastatic melanoma, and it changes during the course of anti-PD-1 therapy. Exosomes may also transfer mRNAs that can be translated into proteins, as well as regulatory miRNAs [178]. In this regard, modifications of exosome PD-L1 mRNA have been described in cohorts of melanoma and NSCLC patients during treatment with anti-PD-1 antibodies, such as nivolumab and pembrolizumab [179]. Notably, a significant increase of exosome PD-L1 transcript was found in patients with a disease progression, suggesting that monitoring of exosome PD-L1 expression may provide useful information on the response to ICIs treatment. Emerging results also indicate that exosomes embedded miRNAs are involved in different pathological processes, such as lung cancer cell proliferation and migration, angiogenesis, and progression [180] as well as in the clinical response to immunotherapy. Recently, Peng and colleagues performed a miRNA sequencing of plasma exosomes from 30 NSCLC patients subjected to immunotherapy. The authors identified three members of the miRNA-320 family (miRNA-320b, miRNA-320c and miRNA-320d) as potential biomarkers for predicting the efficacy of immunotherapy in advanced NSCLCs. In addition, the authors showed that miRNA-125b-5p was downregulated in the post-treatment exosome plasma compared to baseline samples of the patients experiencing a partial response [181].

In conclusion, the previous and other studies address exosomal content such as proteins, mRNAs and miRNAs as promising diagnostic biomarkers and possible therapeutic targets in lung cancer and immunotherapy.

### 4.10. TIS

Currently, a growing body of evidence suggests that the tumor inflamed phenotype correlates with immunotherapy response [182,183]. Consequently, different gene expression signatures related to the tumor microenvironment have been described as predictive markers of clinical benefit in patients undergoing anti PD-1/PD-L1 treatment [171,184,185,186,187,188,189]. Among these, in 2017, Ayers et al. [184] identified an 18-gene profile, namely “Tumor Inflammation Signature” (TIS) by using the NanoString nCounter digital detection technology (NanoString Technologies, Inc., Seattle, WA). The TIS included IFN-γ–responsive genes, related to antigen presentation, chemokine expression, cytotoxic activity, and adaptive immune resistance. By a strict multi-step validation using independent patient cohorts, the authors proposed a TIS score obtained by a weighted combination of 18-TIS gene expression values, able to predict the response across different solid tumors. Recently, Danaher and colleagues explored the immune phenotype using this TIS algorithm on 9000 tumor expression profiles downloaded from The Cancer Genome Atlas (TCGA) [176]. Notably, a high TIS score was mainly associated to tumors with clinical sensitivity to anti-PD-1 blockade, e.g., melanoma, lung tumors, renal carcinoma and cervico-facial cancers. In addition, the TIS score showed a minimal correlation with mutational load in the majority of cancer types. By evaluating the tumor microenvironment, intended as inflamed or non-inflamed phenotype, the authors hypothesized that the TIS score may provide information on the adaptive immune response within the tumor compared to the mutation load, which measures potentially immune activating neoantigen expression.

Overall, these findings underline that gene expression signatures, which assess the tumor inflamed phenotype by simultaneous measures of multiple genes related to the tumor microenvironment, may increase the knowledge of the cancer immune status.

## 5. Discussion

The search for biomarkers able to predict response to immune checkpoint blockade is becoming increasingly important for patient selection. Here, we have presented the most known and emerging biomarkers. Despite promising results, all the reported biomarkers showed limitations in the ability to define a patient as responsive or not responsive to immunotherapy, the most relevant pitfalls being represented by intra-tumor heterogeneity and dynamic changes over time of such biomarkers. More specifically, the different expression of some of these biomarkers within the same lesion, or between the different lesions of a single patient makes biopsy collection relatively unreliable for response prediction, while the possible change in expression of some biomarkers over time raises the question whether repeating assessments of the same predictor several times across the history of a single patient might be useful to identify the ideal time to start ICI.

Another critical issue associated with biomarker analysis in NSCLC is represented by the difficulty of collecting adequate tissue samples, as the relative scarcity of bioptic material obtainable with standard techniques makes the evaluation of tissue-based biomarkers even more complex. Liquid biopsy through blood collection is potentially able to overcome these difficulties, and it might also be more representative of the systemic expression of a specific biomarker, compared to tissue biopsy; moreover, peripheral blood is easily accessible, allowing the evaluation of the variation over time. However, in spite of its potential, liquid biopsy is still hampered by some limitations, mostly characterized by its currently lower sensitivity compared to tumor tissue and by its inability to analyze specific histology-based biomarkers, such as TILs.

Furthermore, as the use of ICIs in the first line is rapidly changing the treatment scenario for advanced NSCLC, robust predictive biomarkers might prove critical for therapeutic decisions, especially in the case of reliable negative predictive factors, which may potentially allow to select those patients who do not benefit from the use of an ICI in addition or in place of platinum-based chemotherapy in first line, irrespective of PD-L1 expression. Similarly, patients who are not going to respond to ICI in second line after platinum-based chemotherapy might potentially be appropriate candidates for other therapeutic strategies, such as chemotherapy plus antiangiogenic agents (e.g., docetaxel plus nintedanib).

In our opinion, there are descriptive biomarkers, as the presence, density and clonality of T cell, that could be the most promising predictive biomarkers to the approved treatment. Furthermore, some of the other biomarker described in this review, like STK11, INFγ, PTEN, could be targeted with novel drugs in order to overcome the resistance to ICIs.

Finally, as novel biomarkers are being identified, some of these molecules might represent potential therapeutic targets on their own right, rather than predictors of benefit from PD-1/PD-L1 or CTLA4 ICIs; as an example, this might be the case of B7-H4, for which a specific inhibitor (FPA150) is currently being studied in solid tumors [190]. In conclusion, a more complete knowledge of the tumoral microenvironment could allow a more efficient selection of those patients that really benefit from ICIs, but in the next future, could lead to select the most suitable treatment strategy for patients in order to realize a real precision medicine.

## Figures and Tables

**Table 1 cancers-12-01125-t001:** Characteristics of main studies with efficacy evaluation according to PD-L1 expression in the advanced setting.

Study	Population (*n*)	Treatment	Method	Cut-Off(s)	Main Findings
Borghaei H, 2015 (CheckMate-057) [1]	455 previously treated advanced non-squamous NSCLC evaluable for PD-L1	Nivolumab (vs. docetaxel)	IHC 28-8	TPS ≥ 1%, 5%, 10%	Association with longer OS, PFS, ORR, DOR at all cut-offs (secondary endpoints)
Herbst R, 2016 (KEYNOTE-010) [3]	1034 previously treated NSCLC with PD-L1 at least ≥ 1%	Pembrolizumab (vs. docetaxel)	IHC 22C3	TPS ≥ 1%, 50%	Association with longer OS at all cut-offs and longer PFS only at 50% cut-off (primary endpoints)
Rittmeyer A, 2016 (OAK) [4]	850 previously treated NSCLC evaluated for PD-L1 expression	Atezolizumab (vs. docetaxel)	IHC SP142	TC1/2/3 or IC1/2/3	Association with longer OS at all cut-offs (coprimary endpoint)
Brahmer J, 2015 (CheckMate-017) [2]	225 previously treated squamous-cell lung cancer evaluable for PD-L1	Nivolumab (vs. docetaxel)	IHC 28-8	TPS ≥ 1%, 5%, 10%	Association with longer OS, PFS at all cutoffs but no ORR (secondary endpoints)
Reck M, 2018 (KEYNOTE-024) [18]	305 untreated NSCLC with PD-L1 ≥ 50%	Pembrolizumab (vs. platinum-based CT)	IHC 22C3	TPS ≥ 50%	Association with longer OS (primary endpoints)
Gandhi L, 2018 (KEYNOTE-189) [22]	616 untreated non-squamous NSCLC	Pembrolizumab/platinum/pemetrexed (vs. platinum/pemetrexed)	IHC 22C3	TPS ≥ 1%	Association with longer OS and PFS regardless of PD-L1 (exploratory endpoints)
Paz-Ares L, 2018 (KEYNOTE-407) [20]	559 untreated squamous-cell lung cancer	Pembrolizumab/ carboplatin/paclitaxel or nab-paclitaxel (vs. carboplatin/paclitaxel or nab-paclitaxel)	IHC 22C3	TPS ≥ 1%	Association with longer OS and PFS regardless of PD-L1 (exploratory endpoints)
Socinski M, 2018 (IMPOWER-150) [21]	692 untreated non-squamous NSCLC evaluated for PD-L1 expression	Atezolizumab/carboplatin/paclitaxel/bevacizumab (vs. carboplatin/paclitaxel/bevacizumab)	IHC SP142	TC1/2/3 or IC1/2/3	Association with longer PFS at all cut-offs (secondary endpoint)

NSCLC: non-small cell lung cancer; IHC: immunohistochemistry; TPS: tumor proportion score; OS: overall survival; PFS: progression-free survival; ORR: objective response rate; DOR: duration or response; TC/IC: tumor cells or tumor-infiltrating immune cell.

**Table 2 cancers-12-01125-t002:** Studies and meta-analyses evaluating tissue TMB in advanced NSCLC patients treated with ICIs.

Study	Population (*n*)	Treatment	Method(s)	Cut-Off	Main Findings
Rizvi et al. [34]	16 = discovery cohort 18 = validation cohort	Pembrolizumab	WES	Median observed TMB value (209 non-synchronous mutations)	High TMS was associated with longer PFS, both in discovery and validation cohort (*p* = 0.01 and *p* = 0.006, respectively) Higher proportion of patients with high TMB experienced clinical benefit (73% vs. 13%; *p* = 0.04).
Rizvi et al. [35]	240	Anti-PD-1 alone or in combination with anti-CTLA-4	Targeted NGS (MSK-IMPACT)	Median observed TMB value (7.4 SNVs/Mb)	High TMB was associated with durable clinical benefit independently from PD-L1 expression.
Carbone et al. (CHECKMATE 026) [36]	312	Nivolumab vs. platinum-based chemotherapy	WES	Low: 0–99 mutations Medium: 100–242 mutations High: 243+ mutations	PFS and ORR were significantly higher with nivolumab in the subgroup of patients with high TMB.
Legrand et al. [37]	342	Atezolizumab	FoundationOne CDx assay WES	16 mutations/Mb	High TMB was associated with improved ORR and DoR.
Herbst et al. [38]	1046 (793 from KEYNOTE-042 and 253 from KEYNOTE-010)	Pembrolizumab vs. chemotherapy	WES	175 mutations/exome	high TMB was associated with improved response and survival with immunotherapy but not with chemotherapy.
Hellmann et al. (CHECKMATE 227) [39,40]	299	Ipilimumab-nivolumab vs. chemotherapy	FoundationOne CDx assay WES	10 mutations/Mb	In the population of patients with high TMB, ipilimumab-nivolumab was associated with improved PFS over chemotherapy. However, ipilimumab-nivolumab was associated with improved OS over chemotherapy irrespective of TMB.
Paz-Ares et al. [41]	675 (70 from KEYNOTE-021; 293 from KEYNOTE-189; 312 from KEYNOTE-407)	Platinum-based chemotherapy with or without pembrolizumab	WES	175 mutations/exome	The benefit from addition of pembrolizumab to chemotherapy in terms of PFS, OS, and ORR was independent from TMB.

Abbreviations: CTLA-4, Cytotoxic T lymphocyte antigen 4; DoR, duration of response; NGS, next generation sequencing; OS, overall survival; PD-1, programmed death protein 1; ORR, objective response rate; PFS, progression-free survival; TMB, tumor mutational burden; WES, whole exon sequencing.

**Table 3 cancers-12-01125-t003:** Studies evaluating ctDNA dynamics in advanced NSCLC patients treated with ICIs.

Study	Population (*n*)	Treatment	Method(s)	Definition of Molecular Response	Main Findings
Anagnostou V, 2019 [59]	Metastatic NSCLC (24)	ICIs *	TEC-Seq	Dramatic reduction in ctDNA to undetectable levels	Molecular response was associated with longer PFS (*p* = 0.001) and OS (*p* = 0.008)
	Stage I-III NSCLC (14)	Neo-adjuvant nivolumab			Molecular response was associated with major or partial pathological response
ALCINA [70]	NSCLC (10), UM (3), MSI-high CRC (2)	PD-1 inhibitors	ddPCR, bi-PAP, NGS	ctDNA levels undetectable at 8 weeks	ctDNA detection at week 8 significant prognostic factor for PFS (*p* < 0.001) and OS (*p* = 0.004)
Giroux Leprieur E, 2018 [71]	Stage IIIB/IV NSCLC (15)	Nivolumab	NGS	30% decrease of ctDNA levels at the first tumor evaluation	9% increase of ctDNA at first tumor evaluation correlated with absence of clinical benefit (AUC ROC 0.75), shorter PFS (*p* < 0.001) and poorer OS (*p* < 0.001)
Goldberg SB, 2018 [72]	Metastatic NSCLC (28)	Pembrolizumab	NGS	>50% decrease in VAF from baseline	ctDNA response was associated with longer time on treatment (*p* < 0.001), superior PFS (*p* = 0.03), and superior OS (*p* = 0.007).
Iijima Y, 2017 [73]	Advanced NSCLC (14)	Nivolumab	NGS	Decreased VAF at 2 weeks	Decreased VAF at 2 weeks correlated with tumor response
Li L, 2019 [74]	Stage III/IV NSCLC (12)	Pembrolizumab	NGS	-	Maximum somatic allele frequency (MSAF) changes correlated with tumor response
Passiglia F, 2019 [75]	Stage IV NSCLC (45)	Nivolumab	qPCR	20% increase of cfDNA at 6 weeks used for molecular progression	cfDNA increase >20% at 6 weeks associated with worse OS (*p <* 0.001) and shorter TTP (*p* < 0.001)

* PD-1 inhibitors as single agent (21) or in combination with CTLA-4 inhibitor (1), LAG3 inhibitor (1) or chemotherapy (3). Abbreviations: TEC-Seq, Targeted Error Corrected sequencing; ctDNA, circulating tumor DNA; UM, uveal melanoma; MSI, microsatellite instability; CRC colorectal cancer; ddPCR, droplet-digital polymerase chain reaction; bi-PAP, bidirectional pyrophosphorolysis activated polymerization; NGS, next generation sequencing; AUC, area under the curve; ROC, Receiver Operating Characteristic; VAF, variant allele frequency; qPCR, quantitative polymerase chain reaction; cfDNA, cell free DNA; TTP, time to progression.

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
