# Peer review of "Precision Medicine for NSCLC in the Era of Immunotherapy: New Biomarkers to Select the Most Suitable Treatment or the Most Suitable Patient"

_cancers, 2020, doi:10.3390/cancers12051125_

Round 1

Reviewer 1 Report

The review provides a thorough discussion of established and proposed biomarkers for PD-1/PD-L1 immuntherapy and their available evidence.

Title: The article is focussed on NSCLC, that may better be reflected in the title

Abstract: Reads like a summarized copy of the introduction + the PD-L1 part. Suggest rewrite to cover the whole scope of the review

Introduction: Mostly about PD-L1. Suggest to add a paragraph on the different "philosophies" that lead to biomarker hypothesis, tumor-centric, microenvironment, systemic / immune state. There are some very prominent review articles such as the "cancer immunogram" that aim at capturing the assumed multidimensionality of the problem.

l50: suggest Acknowledged established

l46: suggest variability, "mutability" suggests it might be actively changed by some action

l71: suggest for example resize put into perspective imperfection limited utility

l281-284: convoluted sentence missing at least an "and". Suggest simpler structure.

l512: this work analysed 31 patient but apparently had no control group. Should be regarded with more caution. Suggest a summarizing sentence at the end of section 4.4.

521-23: Too strong statement. There may be very rare cases with concurrent mutations. In general, dominant oncogenes = EGFR, ALK, KRAS remain mutually exclusive. Do not mix with comutations of dominant oncogene + non-dominant oncogene = PIk3CA, STK11, CTNNB1. This whole discussion is IMO not useful to introduce KRAS/STK11 comutations at all.

556: "the presence of mutations resulted independent on TMB score" -> not comprehensible

Author Response

The review provides a thorough discussion of established and proposed biomarkers for PD-1/PD-L1 immuntherapy and their available evidence.

Title: The article is focussed on NSCLC, that may better be reflected in the title

Abstract: Reads like a summarized copy of the introduction + the PD-L1 part. Suggest rewrite to cover the whole scope of the review

Answer:  thank you for your suggestion, we modified the abstract as follows

Abstract: In recent years the evolution of treatments has made it possible to significantly improve the outcomes of patients with non-small cell lung cancer (NSCLC). In particular, while  molecular targeted therapies are effective in specific patient sub-groups, immune checkpoint inhibitors (ICIs) have greatly influenced the outcomes of a large proportion of NSCLC patients. While nivolumab  activity was initially assessed irrespectively of predictive biomarkers, subsequent pivotal studies involving other PD-1/PD-L1 inhibitors in pre-treated advanced NSCLC (atezolizumab within the OAK study and pembrolizumab in the Keynote 010 study) reported the first correlations between clinical outcomes and PD-L1 expression. However, PD-L1 could not be sufficient on its own to select patients who may benefit from immunotherapy. Many studies have tried to discover more precise markers complementary derived from tumor tissue or from peripheral blood. This review aims to analyze any characteristics of the immunogram that could be used as a predictive biomarker for response to ICIs. Furthermore, we describe the most important genetic alteration that might predict the activity of immunotherapy.” Introduction: Mostly about PD-L1. Suggest to add a paragraph on the different "philosophies" that lead to biomarker hypothesis, tumor-centric, microenvironment, systemic / immune state. There are some very prominent review articles such as the "cancer immunogram" that aim at capturing the assumed multidimensionality of the problem.

Answer: Thank you for your suggestion. We modified the introduction adding

 “ Understanding the complex interaction between the immune system and cancer molecular biology could lead to the definition of a comprehensive schema that can be useful to drive treatment in single situations.  This review aims at defining the state of the art of biomarkers research to guide ICIs therapy, deepening the current knowledges in terms of approved and experimental molecules, both tissue-derived or circulating.”

l50: suggest Acknowledged established:

Answer: we believe the suggestion is appropriate, so we have changed the text.

l46: suggest variability, "mutability" suggests it might be actively changed by some action

Answer: we believe the suggestion is appropriate, so we have changed the text.

l71: suggest for example resize put into perspective imperfection limited utility

Answer: we believe the suggestion is appropriate, so we have changed the text.

l281-284: convoluted sentence missing at least an "and". Suggest simpler structure

Answer. we believe the suggestion is appropriate, so we have changed the text as follows: “Furthermore, Georgiadis et al. employed a targeted hybrid capture NGS system (Agilent SureSelect XT in-solution hybrid capture system). The custom panel, using a validated cutoff of 5 mutations, targeting the predefined regions of interest across 58 genes, reported that bTMB high, before PD-1 blockage, is predictive for PFS in solid tumor (HR 0.23, p=0.003).

l512: this work analysed 31 patient but apparently had no control group. Should be regarded with more caution. Suggest a summarizing sentence at the end of section 4.4.

Answer:  we believe the suggestion is appropriate, so we have changed the text as follows: “the findings were then approached by the assessment of the Receiver Operating Characteristic (ROC) curve demonstrated a high sensitivity and specificity. As seen above, the different lymphocite ratios have the advantage of being easily calculable, minimally invasive, but have shown a prognostic meaning rather than predictive of response to ICIs. Probably, they could be included within more complex scores of prognosis evaluation as objective data.”

521-23: Too strong statement. There may be very rare cases with concurrent mutations. In general, dominant oncogenes = EGFR, ALK, KRAS remain mutually exclusive. Do not mix with comutations of dominant oncogene + non-dominant oncogene = PIk3CA, STK11, CTNNB1. This whole discussion is IMO not useful to introduce KRAS/STK11 comutations at all.

Answer: we believe the suggestion is appropriate, so we have changed the text as follows: “Recent studies have demonstrated, through gene expression profiling, that KRAS mutations are usually but notalways  mutually exclusive with EGFR mutations or EML4-ALK rearrangement [143][144] [35]  and rarely can co-exist with other genomic alterations[145]”

556: "the presence of mutations resulted independent on TMB score" -> not comprehensible

Answer: thank you for your suggestion, there was an error that we correct: ”the presence of mutations resulted independent of PD-L1 expression”

Reviewer 2 Report

This article provides an overview of biomarkers associated with immune checkpoint inhibitors (ICIs) efficacy. The authors have performed a literature search and reported results from most recent studies on selected biomarkers. In some parts of the text, results from these studies have been only partly outlined and the potential impact of novel biomarkers in the clinical setting has not been extensively discussed. The "Tables", I presume, are not correctly ordered and numbered. In the conclusions, authors should provide more substantial personal view and future perspectives on the use and impact of biomarkers in the clinical setting. 

Comments for each paragraph:

- In the abstract, authors should provide a more detailed description about the content of the article, just to excite curiosity of readers.

- Page 3: line 128: you forgot the number of "Table".

- Page 5, at the end of the paragraph "tissue TMB" , I suggest to include a comment about the different cut-offs and methods used to assess the TMB that could be associated with non univocal results in terms of association with efficacy to treatment. A recent landmark study has been published by Anagnostou V, et al. Nat Cancer 2020. 

- Page 7: line 212: the study cited in ref 47, include patients WHO DID NOT RECEIVE ICIs, so it can not be considered to assess a predictive role of cPD-L1

- Page 7: line 220: this sentence "...for a better interpretation of unclear clinical scenarios."..is not clear. Please provide an explanation of what do you mean in this part of the text.

- Page 7: line 229: how this could be "Table 1"?

- Page 11: line 347: you have already defined the meaning of "TIL" above in the text.

- Page 11: line 354: please provide further explanation of the results from this study (ref. 98) evaluating the combination of CD8+ / PD-L1. CD8 + could be more robust biomarker than PD-L1? The study evaluated a combination startegy. Could RT affect the results ?

- Page 14: line 512: at the end of this paragraph, provide a critical comments on the studies you mentioned, almost retrospectives, and discuss how could be the current or future role of this biomarker: alone? Complementary to other biomarkers?

Conclusions should be reinforced, authors should provide a more personal view about which are the most promising biomarkers that could be available within the next years for clinical practice and whether these could be applicable for ICIs as single agent or for combinatorial treatment strategies.

Author Response

This article provides an overview of biomarkers associated with immune checkpoint inhibitors (ICIs) efficacy. The authors have performed a literature search and reported results from most recent studies on selected biomarkers. In some parts of the text, results from these studies have been only partly outlined and the potential impact of novel biomarkers in the clinical setting has not been extensively discussed. The "Tables", I presume, are not correctly ordered and numbered. In the conclusions, authors should provide more substantial personal view and future perspectives on the use and impact of biomarkers in the clinical setting. 

Comments for each paragraph:

- In the abstract, authors should provide a more detailed description about the content of the article, just to excite curiosity of readers.

Answer: thank you for your suggestion, we modified the abstract as follows “Abstract: In recent years the evolution of treatments has made it possible to significantly improve the outcomes of patients with non-small cell lung cancer (NSCLC). In particular, while  molecular targeted therapies are effective in specific patient sub-groups, immune checkpoint inhibitors (ICIs) have greatly influenced the outcomes of a large proportion of NSCLC patients. While nivolumab  activity was initially assessed irrespectively of predictive biomarkers, subsequent pivotal studies involving other PD-1/PD-L1 inhibitors in pre-treated advanced NSCLC (atezolizumab within the OAK study and pembrolizumab in the Keynote 010 study) reported the first correlations between clinical outcomes and PD-L1 expression. However, PD-L1 could not be sufficient on its own to select patients who may benefit from immunotherapy. Many studies have tried to discover more precise markers complementary derived from tumor tissue or from peripheral blood. This review aims to analyze any characteristics of the immunogram that could be used as a predictive biomarker for response to ICIs. Furthermore, we describe the most important genetic alteration that might predict the activity of immunotherapy.”

- Page 3: line 128: you forgot the number of "Table".

Answer:  Thank you for your suggestion. We correct the text

- Page 5, at the end of the paragraph "tissue TMB" , I suggest to include a comment about the different cut-offs and methods used to assess the TMB that could be associated with non univocal results in terms of association with efficacy to treatment. A recent landmark study has been published by Anagnostou V, et al. Nat Cancer 2020. 

Answer: Thank you for you suggestion we modify the paragraph as follows:

 “The introduction of this test in routine clinical practice is challenging, as it is essential to harmonize TMB approaches to ensure comparable results between different studies. Recently, Anagnostou et al, described a TMB corrective factor which was determined on a large cohort of tumor sample from the genome cancer atlas and then validated on a second cohort of patients treated with ICIs. It was based on different levels of tumor purity. This corrective factor seems to greatly increase the predictive value of TMB in prognostication the outcome in patients treated with ICIs, suggesting that the TMB could be largely underestimated in low tumor purity samples resulting in misclassification of patients with these tumors.”

- Page 7: line 212: the study cited in ref 47, include patients WHO DID NOT RECEIVE ICIs, so it can not be considered to assess a predictive role of cPD-L1

Answer: Thank you for your suggestion: there was an error that we removed. We omitted the reference relative to this paper.

- Page 7: line 220: this sentence "...for a better interpretation of unclear clinical scenarios."..is not clear. Please provide an explanation of what do you mean in this part of the text.

Answer: Thank you for your suggestion. We corrected the sentence as follows: “for a better interpretation of unclear clinical scenarios  in those cases of uncertain progression (pseudo-progression or mixed response).”- Page 7: line 229: how this could be "Table 1"?

Answer: Thank you. It was wrong we correct the number. It is the table 2.

- Page 11: line 347: you have already defined the meaning of "TIL" above in the text.

Answer: Thank you for your suggestion. We correct this and other similar error in the text.

- Page 11: line 354: please provide further explanation of the results from this study (ref. 98) evaluating the combination of CD8+ / PD-L1. CD8 + could be more robust biomarker than PD-L1? The study evaluated a combination startegy. Could RT affect the results ?

Answer: thank you for your suggestion. We modified the paragraph as follows:

 “Takaaki Tokito et al retrospectively analyzed predictive relevance of PD-L1 expression combined with CD8+ TIL density in 74 stage III NSCLC patients receiving concurrent chemoradiotherapy between 1999 and 2013. Four groups of patients were analyzed: CD8+ high/PD-L1 neg; CD8+ low/PD-L1 pos; CD8+ high/PD-L1 pos; CD8+ low/PD-L1 neg. At a median follow up of 53 months, the best outcome in term of PFS and OS was showed by CD8+ high/PD-L1 neg group (median not reached), followed by CD8+ high/PD-L1 pos group with a PFS of 17.6 months and an OS of 35.3 months. On the other hand, the group with the worst results was CD8+ low/PD-L1 pos with a PFS of 8.6 months and an OS of 13.9 months. These results underline the negative prognostic value of PD-L1 and highlight the importance of lymphocyte-tumor interface. It is known that RT increases the expression of PD-L1 and promotes anti-tumor immunity [93], Therefore, it is reasonable to think that, in the era of immunotherapy, the combination of PD-L1 and CD8 expression may have an important predictive response value [94].

AND line 403

“Especially, the combination of CD8+ and PD_L1 expression seems to be more robust biomarker than PD-L1 alone.”

- Page 14: line 512: at the end of this paragraph, provide a critical comments on the studies you mentioned, almost retrospectives, and discuss how could be the current or future role of this biomarker: alone? Complementary to other biomarkers?

Answer: Thank you, we modified the sentence as follow:

“the findings were then approached by the assessment of the Receiver Operating Characteristic (ROC) curve demonstrated a: high sensitivity and specificity were documented testing NK cells number .  As seen above, the different lymphocite ratios have the advantage of being easily calculable, minimally invasive, but have shown a prognostic meaning rather than predictive of response to ICIs. Probably, they could be included within more complex scores of prognosis evaluation as objective data.”

Conclusions should be reinforced, authors should provide a more personal view about which are the most promising biomarkers that could be available within the next years for clinical practice and whether these could be applicable for ICIs as single agent or for combinatorial treatment strategies.

Answer: Thank you for your suggestion we modified the text as follows

“In our opinion there are descriptive biomarkers as the presence, density and clonality of T cell that could be the most promising predictive biomarkers to the approved treatment. Furthermore some of the other biomarker described in this review like STK11, INFγ, PTEN could be targeted with novel drugs in order to overcome the resistance to ICIs.

Finally, as novel biomarkers are being identified, some of such molecules might represent potential therapeutic targets on their own right, rather than predictors of benefit from PD-1/PD-L1 or CTLA4 ICIs; as an example, this might be the case of B7-H4, for which a specific inhibitor (FPA150) is currently being studied in solid tumors[190]. In conclusion, a more complete knowledge of the tumoral microeviroment could allow a more efficient selection of those patients that really benefit from ICIs, but in the next future, could lead to select the most suitable treatment strategy for any patients in order to realize a real precision medicine.”

Reviewer 3 Report

This paper provides a comprehensive overview of various acknowledged and emergent biomarkers of immune checkpoint inhibitors especially in NSCLC. The paper has a strong clinical focus and includes many different clinical trial evaluations of the biomarkers. The paper is written in a clear and concise manner and presents the current state of the art in this field. However, the authors present the different evaluations in a very descriptive manner and in general I feel that summarizing and concluding remarks helping the reader to interpret and compare the different clinical trials is missing. If the authors will address this issue as well as the below mentioned point’s I will recommend it for publication.     

The tissue PD-L1 section describe different clinical studies analyzing the PD-L1 expression as a predictive biomarker. This section is missing more summarizing remarks similarly to the single example in line 71-72. In addition, in the end of this section is described several characteristics which challenges the use of the PD-L1 as a biomarker, however a final concluding remark of PD-L1 as a biomarker is missing.

The section 4.1 T-cell clonality would be improved by including more data describing T-cell clonality from a NSCLC perspective. The authors should include the recent paper from Reuben A el al. 2020 (Nat. Comm.).

The authors quite extensively describe the KRAS and STK11 mutations in NSCLC in general. Although the authors briefly mentions the impact of STK11 mutations on the tumor microenvironment (TME) and PD-L1 expression, it would improve this section to increase focus on the impact of KRAS and STK11 mutations on the TME rather than the more general introduction to the mutations in the first part of the section.

The description of IFN-gamma and IL-6 as biomarkers is rather limited. Perhaps the authors should consider joining and extending these sections under a general cytokine heading. As for IFNg, it plays a pivotal role in immunotherapy response and resistance and thus it would improve this section to include a more comprehensive description of INF-g in relation to eg. mutations in JAK1/2 and decreased IFNg signaling.

  Author Response

This paper provides a comprehensive overview of various acknowledged and emergent biomarkers of immune checkpoint inhibitors especially in NSCLC. The paper has a strong clinical focus and includes many different clinical trial evaluations of the biomarkers. The paper is written in a clear and concise manner and presents the current state of the art in this field. However, the authors present the different evaluations in a very descriptive manner and in general I feel that summarizing and concluding remarks helping the reader to interpret and compare the different clinical trials is missing. If the authors will address this issue as well as the below mentioned point’s I will recommend it for publication.     

The tissue PD-L1 section describe different clinical studies analyzing the PD-L1 expression as a predictive biomarker. This section is missing more summarizing remarks similarly to the single example in line 71-72. In addition, in the end of this section is described several characteristics which challenges the use of the PD-L1 as a biomarker, however a final concluding remark of PD-L1 as a biomarker is missing.

Answer: Thank you for your suggestion , at the end of the paragraph we modified the text as follows : “In conclusion PD-L1 seems to be a robust biomarker with several limitation. As we are going to say below, other biomarkers could optimize its performance.”

The section 4.1 T-cell clonality would be improved by including more data describing T-cell clonality from a NSCLC perspective. The authors should include the recent paper from Reuben A el al. 2020 (Nat. Comm.).

Answer: Thank you very much for this suggestion, we included this paper as follows (line 426-432) “Recently, a paper by Reuben et al has described tissue t-cell repertoire in localized NSCLC. Their findings suggested a positive relation between T cell density and clonality. Furthermore, tumor with high PD-L1 demonstrated high T cell density and clonality. Finally, they evaluated the correlation between TMB and T cell clonality. They found that high TMB correlated with high T cell clonality. Indeed EGFR mutant NSCLC presented a lower T cell clonality, that could be a possible explanation of the lower activity of ICIs in these patients [104].”

The authors quite extensively describe the KRAS and STK11 mutations in NSCLC in general. Although the authors briefly mentions the impact of STK11 mutations on the tumor microenvironment (TME) and PD-L1 expression, it would improve this section to increase focus on the impact of KRAS and STK11 mutations on the TME rather than the more general introduction to the mutations in the first part of the section.

Answer: Thank you for the suggestion, we modified the text as follows: “Koyama et al demonstrated that the co-mutation of STK11 resulted in accumulation of neutrophils with T-cell–suppressive effects, and an increase in the expression of T-cell exhaustion markers and immunosuppressor cytokines in mouse model of KRAS-driven NSCLC. Furthermore, the number of TILs was also reduced. Finally, It was described a reduced expression of PD-1 ligand PD-L1 [158]”

The description of IFN-gamma and IL-6 as biomarkers is rather limited. Perhaps the authors should consider joining and extending these sections under a general cytokine heading. As for IFNg, it plays a pivotal role in immunotherapy response and resistance and thus it would improve this section to include a more comprehensive description of INF-g in relation to eg. mutations in JAK1/2 and decreased IFNg signaling.

Answer: Thank you for your suggestion, we modified the text as follows: “IFN-g binding its receptor lead to phosphorylation of JAK 1 and JAK2 and, then, to the dimerization of STAT1-2. These dimers accumulate in the nucleus where they act as trascription factors via IRF1 that finally increase the surface expression of PD-L1.[162] [163].

Reviewer 4 Report

This manuscript by Rossi and co-workers, entitled “Precision medicine in the era of immunotherapy: biomarkers to select the most suitable treatment or the most suitable patient?”. described the most recent data about the potential biomarkers for response/resistance to ICI.
This review is very interesting and well written, with a full exploration of the recent data above all on NSCLC.
I suggest adding only few data in the review, in order to obtain a complete overview of this topic in NSCLC.
Major points.
1. The authors described the role of sPD-L1 as potential biomarker. Concerning the role of sPD-L1, I think that it could be relevant adding data about the ability of chemotherapy to modify sPD-L1. As in the first line setting pembrolizumab is associated with platinum-based chemotherapy in NSCLC adenocarcinoma subtype (platinum plus pemetrexed), it could be relevant if chemotherapy modify sPD-L1 in all the patients or only in patients responsive to ICI. This could be useful to select patients responsive to combined treatment.
2.  Moreover, it could be of particular interest the addiction of a small section focused on the potential role of exosomes (and their content) as potential biomarker in liquid biopsy.
3. In the chapter 4 the authors described a lot of possible biomarkers for sensitivity or resistance to ICI. I think that the authors should add a paragraph concerning the role of JAK/STAT mutations as a possible mechanisms of resistance to ICI treatment.
Minor points.
The authors should remove semicolons in the manuscript between the references (for example lane 465, between the refs. 125 and 126; lane 523, between the refs. 141, 142and 35; lane 532-534, refs. 146, 147 and 148.
Lane 544: please add “it” at the sentence “…. has been suggested…”.

Author Response

This manuscript by Rossi and co-workers, entitled “Precision medicine in the era of immunotherapy: biomarkers to select the most suitable treatment or the most suitable patient?”. described the most recent data about the potential biomarkers for response/resistance to ICI.
This review is very interesting and well written, with a full exploration of the recent data above all on NSCLC.
I suggest adding only few data in the review, in order to obtain a complete overview of this topic in NSCLC.
Major points.
1. The authors described the role of sPD-L1 as potential biomarker. Concerning the role of sPD-L1, I think that it could be relevant adding data about the ability of chemotherapy to modify sPD-L1. As in the first line setting pembrolizumab is associated with platinum-based chemotherapy in NSCLC adenocarcinoma subtype (platinum plus pemetrexed), it could be relevant if chemotherapy modify sPD-L1 in all the patients or only in patients responsive to ICI. This could be useful to select patients responsive to combined treatment.

Answer: Thank you for your suggestion, we modified the text as follows:” Interestingly, some data suggest that first line chemotherapy (eg platinum and pemetrexed) could significantly increase sPD-L1 median level from baseline, while targeted therapies did not. [47]”

  1. Moreover, it could be of particular interest the addiction of a small section focused on the potential role of exosomes (and their content) as potential biomarker in liquid biopsy.

Answer: Thank you for your suggestion, we added a paragraph on the potential role of exosome: “4.9. Exosomes

Exosomes are extracellular vesicles with a size ranging from 40 to 100 nm. These small vesicles, released by an endosomal maturation, carry biomolecules such as genetic material, including DNA, mRNAs, miRNAs and other noncoding RNAs, as well as proteins[172]. Exosomes can be found in various body fluids, such as the blood, sputum, urin, or malignant effusions, and they play a pivotal role in intercellular communication [173]. Interestingly, tumor-derived exosomes bearing their molecular and genetic cargo may reflect the composition of tumor cells. A growing body of evidence indicates that tumor-derived exosomes can rescue tumor cells by evading the surveillance of immune cells, which could represent a therapeutic target[174] . Tumor-derived exosomes in fact, can suppress the function of immune cells by transferring their content, thus regulating lung cancer progression [175]. Recent evidence has demonstrated that exosomes from various tumor cells, including lung cancer cells carry immunosuppressive PD-L1 on their surface, which can be up-regulated by IFN-γ, thereby facilitating tumor growth by suppressing the function of CD8+ T cells and the immune response [176,177]. In particular, the level of circulating exosomal PD-L1 positively correlates with that of IFN-γ in patients with metastatic melanoma, and it changes during the course of anti-PD-1 therapy. Exosomes may also transfer mRNAs that can be translated into proteins, as well as regulatory miRNAs [178]. In this regard, modifications of exosome PD-L1 mRNA have been described in cohorts of melanoma and NSCLC patients during treatment with anti-PD-1 antibodies, such as nivolumab and pembrolizumab [179]. Notably, a significant increase of exosome PD-L1 transcript was found in patients with a disease progression, suggesting that monitoring of exosome PD-L1 expression may provide useful information on the response to ICIs treatment. Emerging results also indicate that exosomes embedded miRNAs are involved in different pathological processes, such as lung cancer cell proliferation and migration, angiogenesis, and progression [180] as well as in the clinical response to immunotherapy. Recently, Peng and colleagues performed a miRNA sequencing of plasma exosomes from 30 NSCLC patients subjected to immunotherapy. The authors identified 3 members of the miRNA-320 family (miRNA-320b, miRNA-320c and miRNA-320d) as potential biomarkers for predicting the efficacy of immunotherapy in advanced NSCLCs. In addition, the authors showed that miRNA-125b-5p was downregulated in the post-treatment exosome plasma compared to baseline samples of the patients experiencing a partial response [181].

In conclusion, the previous and other studies address exosomal content such as proteins, mRNAs and miRNAs as promising diagnostic biomarkers and possible therapeutic targets in lung cancer and immunotherapy.”

  1. In the chapter 4 the authors described a lot of possible biomarkers for sensitivity or resistance to ICI. I think that the authors should add a paragraph concerning the role of JAK/STAT mutations as a possible mechanisms of resistance to ICI treatment.

Answer: Thank you for your suggestion, we modified this sentence as follows: “. IFN-g binding its receptor lead to phosphorylation of JAK 1 and JAK2 and, then, to the dimerization of STAT1-2. These dimers accumulate in the nucleus where they act as trascription factors via IRF1 that finally increase the surface expression of PD-L1.[162] [163].”

Minor points.
The authors should remove semicolons in the manuscript between the references (for example lane 465, between the refs. 125 and 126; lane 523, between the refs. 141, 142and 35; lane 532-534, refs. 146, 147 and 148.

Answer: Thank you, we correct the text.
Lane 544: please add “it” at the sentence “…. has been suggested…”.

Answer: Thank you we correct the text.

Round 2

Reviewer 2 Report

The authors have answered to my comments and addressed some of my
concerns sufficiently. They also included some more information,
following my suggestions.